# Integrating a New Dietetic Care Process in a Health Information System: A System and Process Analysis and Assessment

**DOI:** 10.3390/ijerph19052491

**Published:** 2022-02-22

**Authors:** Gabriele Gäbler, Deborah Lycett, Walter Gall

**Affiliations:** 1Department of Science, Research and Development, Austrian Association of Dietitians, Grüngasse 9, 1050 Vienna, Austria; 2Centre for Intelligent Healthcare, Coventry University, Priory Street, Coventry CV1 5FB, UK; ab5042@coventry.ac.uk; 3Center for Medical Statistics, Informatics and Intelligent Systems, Medical University of Vienna, Spitalgasse 23, 1090 Vienna, Austria; walter.gall@meduniwien.ac.at

**Keywords:** standardized terminology, International Classification of Functioning, Disability, and Health (ICF)-Dietetics, nutrition and dietetics, clinical documentation, health information system, electronic health record

## Abstract

Managing routinely collected data in health care and public health is important for evaluation of interventions and answering research questions using “real life” and ”big data”. In addition to the technical requirements of information systems, both standardized terminology and standardized processes are needed. The aim of this project was to analyse and assess the integration of standardized terminology and document templates for a dietetic care process (DCP) into the health information system (HIS) in a hospital in Austria. Using an action research approach, the DCP was analysed through four expert interviews and the integration into the HIS through two expert interviews with observations. Key strengths and weaknesses for the main criteria (“integration of the ICF catalogue”, “adaption of the document templates”, “adaption of the DCP”, and the “adaption of the user authorizations”) were presented and proposals for improvement given. The system and process integration of the DCP is possible, and the document templates can be adapted with the software currently in use. Although an increase in resources and finances required is to be expected initially, the integration of a standardized dietetic terminology in combination with a standardized process is likely to improve the quality of care and support outcomes management and research.

## 1. Introduction

Managing routine outcome data is essential for auditing and benchmarking, evaluating current interventions and conducting pragmatic research on big data in health care and public health [1,2]. The use of a standardized care process and standardized terminology for the documentation of individual therapy and population-level interventions provides clear, unambiguous terms to support the comparability and management of routinely collected data [3,4]. 

This is supported by the World Health Organization (WHO) in their recommendation to use the International Classification of Functioning, Disability, and Health (ICF), a multipurpose classification for various disciplines describing the functional state of health and its contextual factors [5,6]. Use of the ICF is recommended in combination with the International Classification of Diseases (ICD) [7] and the International Classification of Health Interventions (ICHI) [8,9]. Together, these capture the situation of patients at a given point in time and cover the most important parts of the health system [5,10]. They are a reference system supporting clinical practice and data collection at the individual (micro), institutional (meso) and social (macro) level [8,11] of the health system. The ICF is considered a third health indicator (in addition to mortality and morbidity) that can be used to monitor the achievement of health strategies at all these levels [12].

Numerous articles can be found regarding ICF. These describe how the ICF framework can be used in multidisciplinary health care [13,14,15,16,17,18] and discuss the development and validation of ICF Core Sets (sets of ICF categories relevant for patients with a certain heath condition) to facilitate multidisciplinary assessment [19,20,21,22,23,24,25,26,27]. Articles compare the content of instruments to measure functioning of patients [28,29,30,31,32,33,34,35] and describe the linking of problems experienced from a patient perspective in daily life [36,37,38]. Furthermore, studies discuss the implementation of ICF-based tools in clinical practice [39,40,41,42,43,44,45,46] and ICF use in electronic health records (EHRs) [47,48]. 

Although the ICF is used by various health professions, e.g., physicians, nurses, physiotherapists, occupational therapists, and speech therapists, it has not yet been used by dietitians (dietitians are a legally recognized health profession and in Austria belong to the Higher Medical-Technical Professions (MTDs) [49]). The ICF was lacking in dietetics-related categories and unsuitable for the documentation of the dietetic care process (DCP) until the Dutch Association of Dietitians, in collaboration with the Dutch Institute of Allied Health Care, expanded the ICF with approximately 900 dietetics-related categories [50]. This ICF-Dietetics is part of the Classifications and Coding Lists for Dietetics (CCD) in the Netherlands [51]. The CCD was developed as a standardized dietetics terminology and consists of various classifications and coding lists, e.g., a classification for interventions modified by the ICHI and a list of goals [51]. The second standardized dietetics terminology applied worldwide is the Nutrition Care Process Terminology (NCPT) developed by the American Academy of Nutrition and Dietetics, which is exclusively a nutrition and dietetics terminology [51]. Beginning with an action research process, the ICF-Dietetics was compared with the NCPT, translated, validated, and clinically pre-tested to develop an Austrian–German ICF-Dietetics version [52]. Document templates were developed alongside this for the DCP, based on ICF documentation tools [13,53]. A multicentre study [50] and a focus group study [54] showed that the integration of the ICF-Dietetics into the DCP was possible, and an Austria-wide implementation of the ICF-Dietetics was advocated. However, a multifaceted implementation strategy is required, with the integration of ICF-Dietetics into existing health information systems (HIS) seen as a prerequisite for a successful implementation [54]. 

HIS that support both structured and free text information can facilitate the collection, presentation, sharing, and use of health information [47]. In this context, standardized terminologies contribute to improving the quality and comparability of data [4,47]. The ICD of the WHO classifications is already automatically integrated into the standard software of Austrian HIS. Furthermore, the integration of the ICF into a HIS has already been explored in several studies [4,47,55] as mentioned above, but standard implementation for clinical practice is not yet available [47]. 

The implementation of standardized terminology in a HIS requires a system analysis and system assessment as a first step [55,56,57]. This allows obstacles and potential for improvements to be identified, timely adaptations, and, if necessary, an alternative proposal to be made [56,58]. In hospitals, HIS are used by different providers, have different requirements, and must meet different challenges. This heterogeneity makes it necessary for this system analysis and assessment to be carried out in the context of implementation projects based on a specific HIS. Principles from learning here can be applied to other settings.

The aim of this project was therefore to analyse and assess the integration of a new DCP using standardized terminology and templates for documentation in a HIS with an existing service request called “dietological council” in a large hospital in Austria.

## 2. Materials and Methods

This study reports on a system and process technology analysis and assessment focussing on integrating a DCP into an existing HIS. This was conducted in two phases: first, analysis and assessment of the new DCP, and second, analysis of current process of dietetic referral (“dietological council”) and assessing the integration of the new DCP. Using an action research approach [59,60,61] with qualitative semi-structured interviews facilitates identifying strengths and weaknesses of the system and process integration. 

The DCP was developed by an expert task force based on national and international literature and using an interactive development process. It is described in detail elsewhere [62]. The clinical practicability and applicability of the ICF-Dietetics and the templates of the current study have already been evaluated earlier in nutrition and dietetic practice by means of a pre-test and focus group study in a hospital in Austria. The General Hospital of Vienna was chosen for the system and process technology analysis and assessment, as the dietitians agreed that the ICF-Dietetics would be useful and acceptable for clinical practice. They also stressed the need to integrate the new process with the document templates and standardized terminology in the HIS.

### 2.1. Analysis and Assessment of the New DCP

Four interviews were conducted by a computer scientist (principal investigator) with an academic dietitian recruited from the developers of the DCP. The interviews were structured around the five sub-processes of the DCP (dietetics assessment, dietetics diagnosis, dietetics goal setting, dietetics intervention, and dietetics evaluation). These interviews were part of an iterative development process whereby the four typical development phases of planning, doing, checking, and acting (PDCA-cycle) were performed several times in order to improve the quality of the analysis results continuously [59,60]. Feedback from other developers of the DCP and the interviewer also played a part. During this process, graphical representations of the new DCP and a precise description for the analysis and assessment with respect to its five sub-processes were carried out by the principal investigator. The purpose of these was to develop, test, and refine the process through discussion of how it should work, of what works well and what barriers arise. The graphical representation of the new DCP with reference to the document templates and the standardized terminology (classification catalogues) was depicted in a simplified process diagram (Figure 1). This is described in detail in Appendix A. (The referral required according to § 2 of the MTD law [49] in case of a medical diagnosis is not shown explicitly in Figure 1, since the focus is on the implementation of the new DCP by dietitians, and this starts with the assessment. Expert systems and clinical decision support were hypothetically included in the interviews (Appendix A), but were not assessed).

The ICF-Dietetics is the only classification catalogue available for implementation at the time of the project. Further catalogues for nutrition-related problems and nutrition related goals are planned and therefore were already taken into account in the description of the new DCP. Similarly, the planned classification catalogue for interventions for dietetics was integrated and referred as ICHI-Dietetics, because it should be based on the ICHI [9] and adopted for dietetics.

### 2.2. Analysis of Current System of Dietetic Referral (“Dietological Council”) and Assessing the Integration of the New DCP 

Two further interviews, together with on-site observations of use of the current process, were conducted by the principal investigator with the leading dietitian of the hospital. These interviews also formed part of an interactive process with feedback from other dietitians of the hospital. During this development process, the current process was visualized as a business process model (“extended event-driven process chain” (eEPC) [64,65]) and used as a framework for assessing the integration of the new DCP within it. The current process is divided into four sub-processes due to its complexity (Figure 2). The whole business process model is shown and described in Appendix A. The following specific requirements were applied to the modelling language of the current process: 1. Trivial notation elements for users without modelling expertise, kept as general as possible and representable without additional specifics or explanations; 2. Modelling language close to the existing enterprise resource planning (ERP) software SAP, and standardized modelling language that enables processing by external persons. The software “EdrawMax” [66] was used for modelling.

Qualitative data of the assessment was mapped against four main criteria that are important for the integration of a new process with documents and standardized terminologies: for the system integration, the “integration of the ICF catalogue” and the “adaption of the document templates”, and for the process integration, the “adaption of the new DCP” and the “adaption of the user authorizations”.

## 3. Results

The key strengths and weakness of the system and process technology integration (“integration of the ICF catalogue”, “adaption of the document templates”, “adaption of the new DCP”, and the “adaption of the user authorizations”) are summarised in Table 1 and explained below with examples of proposals for improvement. Further detailed descriptions and suggestions for improvement can be found in Appendix A.

### 3.1. System Integration Aspects (Integrating the New DCP into the Current Process ‘Dietological Council’)

The service request “dietological council” of the current process uses the HIS with an EHR to create the documents. The implemented HIS software is SAP [67] with the modules “IS-H” [68] and “i.s.h.med” [69]. Dietitians are fully integrated in both modules. No additional software is required. The system activities for dietitians include access to patient data, changes of “service requests”, performance indicator entry, and documentation. The written documents use parameterized documents (documents that have already specified certain parameters, values, or variables) with free text and text templates.

A change or creation of the documents according to the criteria of the DCP is possible regardless of the implementation of the classification catalogues and only requires system-technical settings (customizing) in the “i.s.h.med”. The technical integration includes the software adaptations for the implementation of the ICF-Dietetics and other classification catalogues as well as document templates for documenting the DCP.

#### 3.1.1. Integration of Classification Catalogues for the New DCP 

At present, neither a standardized dietetics terminology nor a general classification catalogue containing nutrition-related concepts is used. However, classification catalogues, such as the ICD-10 catalogue, are already implemented in the existing SAP “IS-H” and “i.s.h.med” systems. The only available classification catalogue for implementation of the new DCP was the ICF-Dietetics. However, the specifications for the use of the entire classification catalogues allow individual gradual adoptions, so ICF-Dietetics integration is seen as just the first step. The ICF-Dietetics is integrated into the HIS by importing the contents of the Excel worksheet into a created SAP table. This is based on the existing ICD-10 template and builds in its organizational and process-related requirements such as the validity date (start and end date). The ICF-Dietetics input file contains five columns, of which only the first (codes) and the second column (title of the category) will appear on the electronic schedule. The remaining three columns (description, inclusion, and exclusion) are only visible when the search or coding input help is activated. Using this input help, the search for texts can be carried out in all five input fields. An ontological search is not possible. The technical adaptation options of the existing Excel format and the trivial technical structure facilitate the implementation of the ICF-Dietetics. Thus, the current system meets the requirements for the technical implementation of ICF-Dietetics in Excel format and other catalogues with the same structure. To increase the user-friendliness and ease of work, there is the need for various support tools for users (e.g., search aids, core sets, keyword catalogue).

#### 3.1.2. Adaptation of the Document Templates for the New DCP

Four documents are planned for the new DCP (Appendix A). The documents for the current process are based on the more precise SAP technology of the parameterized documents of the “i.s.h.med”. The parameterized document can access all data in the SAP system so that the patient data can be fully integrated. In addition, parameterized documents can also take over data from other parameterized documents. The ICF-Dietetics catalogue entries (codes and titles) migrated into the SAP system as table contents can thus be activated for documentation in a document. The creation of the parameterized documents does not require any additional software, programming, or licenses. The documents can be created using the future terminology based on the new DCP. The layout specifications (such as font, font size, and patient header) can be adopted from the existing diagnostic document. The structural changes (content according to the bio-psycho-social model of ICF-Dietetics) are to be implemented in the “body” (textual information) of the document. Since the description of the functional state of health is based on ICF-Dietetics and its bio-psycho-social model, it restricts the documentation templates to this model. A combined use of the ICF catalogue with the NCPT catalogue for nutrition diagnosis (nutrition problems) and interventions would, however, be possible without changing the document templates.

### 3.2. Process Integration Aspects (Integrating the New DCP into the Current Process)

The current process is subject to internal work instructions for therapeutic health professionals of the MTD group. These work instructions describe a general therapeutic process from the start (dietetics assessment of initial situation) to the end of therapy (evaluation). For the proper implementation of the new DCP, the document templates and classification catalogues must be used from the beginning of the process (the dietetics assessment). For this purpose, the dietitian must be provided with a portable device (e.g., laptop or tablet) that has to be used for the “dietological visit” (the patient consultation) in the future. Due to data entry, conversation with the patient can be disrupted during the visit; additionally, expenditure of work and time may be increased. In this regard, an increase in resources and financial expenditure is to be expected. In the long term, however, the new DCP is to be seen as an advantage due to an efficient and effective workflow and the possibility of outcomes research.

In general, the document templates were created without administrative patient data (header) such as name, date of birth, date of admission. However, this information is important regarding document security and clear patient identification. The structure of the header will be created according to the specifications of the respective hospital administration and the minimum necessary for the performance requirements.

#### 3.2.1. Adaption of the New DCP

The new aspects of the DCP are mainly due to the integration and use of new document templates and standardized terminology and not a change in the process flow, as shown in the eEPC framework (Appendix A). The potential for improvement primarily relates to the document templates (e.g., the documentation of dietetics diagnosis and goal setting), as described in Appendix A. The implementation of the new features of the DCP requires the creation of at least two electronic documents, the “ICF-Dietetics Assessment Sheet” and the “ICF-Dietetics Goal Setting Sheet”.

If several dietological services are carried out during an inpatient hospital stay, many dietological documents must be created. This can lead to difficulties and misunderstandings, especially if related dietological documents contain different creation dates. A reference on the created documents to the associated documents would simplify the problem. This can be done by referring to the creation date of the associated documents or an ID (e.g., referral ID) on the dietological document. The reduction in the number of documents per care process can improve the overview of the dietological documents in the intra- and interdisciplinary area and makes administration easier. One possibility for simplification would be the illustration (output file) of the therapy report and the subdivision of the document into the respective documents of the sub-process using separate tabs. The new DCP depicted here does not contain a written document that can be given to the patient as a final report or therapy report. It is recommended in any case to integrate a semi-automatic final report as summary of important information, e.g., dietetics diagnosis, the goals, and the interventions.

#### 3.2.2. Adaption of the User Authorizations

The existing authorizations allow dietitians to carry out their process without restrictions. For integration of the new DCP with new templates and standardized terminologies, additional administrative data (e.g., living situation) are required for dietetics’ diagnosis formulation, which could go beyond the current system-technical settings. A system-based authorization adaptation would have to be implemented as well as an evaluation in terms of data protection. The new DCP may require adaptations in the documentation. When data is entered by the dietitian (e.g., weight), it must be checked whether the data entered are only saved and visible in the dietological document or are generally made available in the EHR of the HIS.

The input of medical data into a HIS is carried out by the various health professionals. It must be noted whether certain patient data from other professional groups are also documented in the HIS or on paper. If medical patient data (e.g., weight) is saved in the EHR and not directly in the document of dietitians, the respective implementation authorizations (saving, changing, and deleting) for the respective areas must be checked and structured.

## 4. Discussion

This work demonstrates a system and process analysis and assessment for the integration of a new therapeutic process, the DCP, with a standardized terminology and document templates in an existing HIS. The study used an action research approach. The implementation of the DCP and standardized terminology in HIS will support the documentation of the individual care process and thus quality assurance, e.g., transparency and continuity of care [3,70], on the one hand and facilitate the comparison of data and thus outcomes management and research on the other hand. As a result, best practice methods could be established and the effectiveness of care improved [3]. Moreover, no study was found in the literature that describes the analysis and assessment of the integration of a therapeutic process with standardized terminology. Thus, our detailed description, assessment, and proposals for improvement can assist others in integrating the DCP and the ICF-Dietetics in their HIS. Our study can therefore be of interest to other institutions in Austria as well as in other countries. 

However, we want to emphasize that the change of a process must also take into account the non-technical aspects. During an information system analysis and assessment, it is important that the people involved in the process with their roles and skills are considered accordingly. Therefore, established process models for system analysis and assessment use two dimensions to categorize the areas of the criteria, namely, organizational aspects and the technical aspects of information tools [55,57]. Our system and process analysis and assessment build on a previous pre-test and focus group study, which evaluated the clinical practicability and applicability of the new DCP with the ICF-Dietetics and the templates as a “paper and pencil” version [54]. For this reason, we focused our findings to four main criteria which are important for the integration of a new process with documents and standardized terminologies: “integration of the classifications catalogue”; “adaption of the document templates”; “adaption of the new DCP”; and the “adaption of the user authorizations”. Our results showed strengths and weaknesses in each of these areas; these were consistent with other literature as described below, and obstacles were considered surmountable. In accordance with action research [59,60], the implementation of the new process will be evaluated in an implementation project and will be described elsewhere.

### 4.1. Integration of the ICF Catalogue

A great advantage of the ICF-Dietetics is that it is based on a main classification of the WHO that can be used in multidisciplinary patient management [47,54]. A disadvantage is that an ontological search within ICF-Dietetics as in “DIACOS” [71] is not possible. Maritz et al. [47] and Bales et al. [72] also mention two disadvantages of the ICF: first, that it does not correspond to all features of a formal terminology (it is not yet interoperable), and second, that an ontology is not yet available. However, the “ICF Practice Manual” [73] states that an ontological presentation of the ICF has been in progress since 2008 in order to be compatible with e-health systems. Two aspects are explicitly mentioned in order to facilitate the integration of the ICF in EHRs: first, the formalization of the knowledge representation in ICF (ontology development), and second, the establishment of linking with other clinical terminologies (e.g., SNOMED CT). Both aspects will be important for the application of ICF-Dietetics in the future. Health professionals need a wide range of clinical and administrative patient data available in EHRs for their daily work. To support health professionals in sharing EHRs, integrated communication by means of vocabularies such as SNOMED CT or LOINC is crucial [4]. Therefore, in the future, mappings to other terminologies, especially to SNOMED CT, will enable the international exchange of ICF data in standardized data models [74,75,76] and facilitate big data analysis [77]. To this end, linkage to other established vocabularies is an important goal in addition to establishing a uniform framework and language in dietetics.

It can be assumed that standardized terminologies will become indispensable for the fulfilment of quality criteria in the health sector, such as patent orientation, transparency, effectiveness, and efficiency [78,79]. This implies that the optimization and further development of these terminologies will take place quickly. Experience in the application of ICF/ICF-Dietetics shows that due to the complexity and number of categories (approx. 2400), assigning the optimal category (coding) is found to be difficult and time-consuming [54,80]. To increase the user-friendliness and ease of work, there is the need for various support tools for users (e.g., search aids, core sets, keyword catalogues). These support tools are essential aspects for the integration of ICF-Dietetics [81,82,83]. They increase acceptance and correct coding by employees. 

Finally, it should be mentioned that to meet quality criteria using standardized terminology, the implementation of a new terminology is accompanied by interactive training courses on motivation, professional coding, and use [54,79,83].

### 4.2. Adaptation of Document Templates 

Document templates can improve the consistency and completeness of the documentation, but they can also take more time [4,84]. Studies have shown that precise, unrestricted free text entry is most effective for communicating and coordinating complex tasks [85]. For the dietetics goals, both the ICF-Dietetics catalogue and another special goal catalogue should be used, and free text should also be possible. However, free text in classification catalogues generally contradicts its use in electronic patient files [4]. In contrast, structured data fields are required to search and process data electronically, e.g., to evaluate the effectiveness of interventions, for research, or for administrative purposes [85]. 

### 4.3. Adaptation of New DCP

The new DCP is in line with the legal guidelines in accordance with the FH-MTD training regulation [86] and the MTD law [49] in Austria. It can be seen as a detailed guide for dietitians on how to provide and uniformly document care using standardized terminologies. Due to the specific procedure and the use of the classification catalogues, this new DCP contributes to the quality improvement of clinical dietetic practice, especially in terms of implementation and transparent documentation [47,70,87,88,89]. It increases personalized patient-oriented treatment and enables better decision making in goal setting and the selected interventions [2,3]. A multidisciplinary terminology, such as the ICF, enables efficient and effective work in intra- and interprofessional areas [4,17,47,55]. In addition, the standardized procedure and the use of standardized terminologies enable the evaluation and comparison of outcomes [2,3,88].

### 4.4. Adaption of the User Authorizations

Due to the interdisciplinary character of a dietological service (referral by physician, service allocation by administration), adjustments can be necessary not only in dietetics, but also across departments. Entering patient data from various health professionals into the system without precise regulation can lead to redundancies and, under certain circumstances, to misunderstandings and misinterpretations. In our study, we analysed organizational aspects other than our technical aspects in terms of roles by means of the eEPC, which provided us with a clear and structured basis for the analysis and assessment and, therefore, can be recommended for other projects.

### 4.5. Limitation of the Work

A system analysis and assessment can only be carried out for one organization and application due to the heterogeneity of HIS. In this respect, these results cannot be directly transferred to other projects, but this project can be seen as an example of the methodological approach. We have focused on four criteria in this work, but a wide range of other criteria could also be important. The technical system settings of the SAP program in the area of dietitians were not taken into account for reasons of operational safety. Possible customer changes in the SAP program could therefore have remained undetected. This can lead to additional work due to implementation modifications. This aspect must be considered in a financial plan for integration. Furthermore, expert interviews may not produce fully objective analysis; however, in combination with feedback during the development process, we were able to increase the quality of the results. Another limitation can be seen that the catalogues were still in development; however, as described above a stepwise integration is possible. Finally, the fact that no evaluation of the implementation of the new DCP was included can be seen as a limitation of this study. In this context, we would like to emphasize our methodological approach in the sense of action research, which enables a step-by-step translation and implementation of knowledge into practice.

## 5. Conclusions

According to the project specifications, the new DCP meets the legal requirements. An integration of it with the standardized terminologies and document templates in the existing HIS is technically feasible. The representation of the current process shows that there are no system-technical obstacles, and the dietetics service request “dietological council” can be carried out with the new features of the DCP. The potential for improvement discussed does not concern the system-technical integration of the process itself but technical support tools for the user-friendliness of the classification catalogues and especially the proposed document templates. Although an increase in resources and finances required is to be expected initially, the integration of a standardized terminology in combination with a standardized process can improve the quality of care in the long term and support outcomes management and research.

## Figures and Tables

**Figure 1 ijerph-19-02491-f001:**
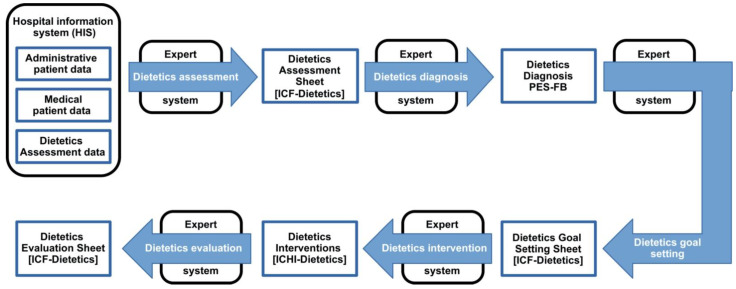
New DCP with document templates and standardized terminology (adapted from Krottenmüller [63]).

**Figure 2 ijerph-19-02491-f002:**
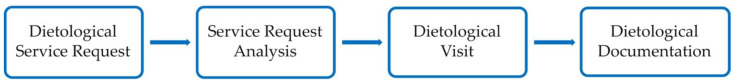
Sub-processes of the current system “dietological council” (adapted from Krottenmüller [63]).

**Table 1 ijerph-19-02491-t001:** Summary of results mapped against the four criteria for the system and process integration: “integration of the classification catalogues”, “adaption of the document templates”, “adaption of the new DCP”, and “user authorizations”.

Current Process	New DCP ^5^	Strength/Facilitators	Weakness/Obstacles
System			
used software is SAP ^1^	software remains SAP no additional software is required	dietitians are fully integrated in the current system	
Integration of the classification catalogues			
no catalogues	ICF ^6^ -Dietetics (available) ICHI ^7^ -Dietetics (in process of planning) nutrition-related problems (available as word file) dietetics diagnosis (in process of planning) dietetics goals (in process of planning)	classification catalogues, such as the ICD-10 ^8^ catalogues, are already implemented in the existing system, and its importing programs can be used as template and be adapted Excel format of the ICF-Dietetics facilitate the implementation	an ontological search within ICF-Dietetics catalogues is not possible not all catalogues are available
Adaption of the document templates			
all information in one document “dietological council” using parameterized documents	four documents: “ICF-Dietetics Assessment Sheet” “ICF-Dietetics Goal Setting Sheet” “ICHI-Dietetics Intervention Sheet” “ICF-Dietetics Evaluation Sheet”	an adaption of the currently used document is possible parameterized documents can take over data from other documents, thus enabling coding	ICF-Dietetics and its bio-psycho-social model restrict the documentation templates of the assessment
Process in general			
applied according to the MTD ^2^ law and internal work instructions the service request “dietological council” uses the HIS ^3^ with an EHR ^4^ to create the documents four sub-processes	corresponds to the MTD law and internal work instructions of the MTDs five sub-processes documentation with new templates and standardized terminologies	enables transparent documentation enables efficient and effective workflow in intra- and interprofessional areas enable the evaluation and comparison of outcomes	increase in resources and finances required, e.g., portable devices complexity of coding support tools are needed document templates generally missing patient data (header)
Adaption of the new DCP			
“Dietological service request”	no corresponding part		
“Service request analysis”	first part of dietetics assessment	automatic transfer of data from EHR is possible	no fields are provided for automatic transfer currently only free text
“Dietological visit”	dietetics assessment “ICF-Dietetics Assessment Sheet” dietetics diagnosis “ICF-Dietetics Goal Setting Sheet” dietetics goal setting “ICF-Dietetics Goal Setting Sheet” dietetics intervention “ICHI-Dietetics Intervention Sheet” part of dietetics evaluation	structured implementation of data collection	type of diagnosis is missing manner of PES-FB ^9^ statement manner of goal setting dietetics aid catalogues and patient letter/final report are missing
“Dietological Documentation”	part of dietetics evaluation		information and findings are documented in several documents
User authorizations			
carry out the process without restrictions	additional administrative data are required	authorizations allow dietitians to carry out their process without restrictions	a system-based authorization adaptation and data protection are required

^1^ Systems Applications Programs. ^2^ Higher Medical-Technical Professions. ^3^ Health Information System. ^4^ Electronic Health Record. ^5^ Dietetic Care Process. ^6^ International Classification of Functioning, Disability and Health. ^7^ International Classification of Health Interventions. ^8^ International Classification of Diseases Version 10. ^9^ Dietetics diagnosis formulation, including the nutrition-related problems (P), the etiology (E) and the symptoms or signs (S) of the problem and according to the ICF coding guidelines the facilitators (F) and barriers (B).

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
