# Peer review of "Integrating a New Dietetic Care Process in a Health Information System: A System and Process Analysis and Assessment"

_ijerph, 2022, doi:10.3390/ijerph19052491_

Round 1

Reviewer 1 Report

The authors address correctly my suggesttions.

However, in my humble  opinion the innovation continue being unclear to me. For me be the first and only system that incorporates ICF in a EHR is not enough to describe the real advances that your system provides, (that probably are a lot). I consider the the authors can improve this part

In any case, I consider that the paper in general  can be published in the journal.

Author Response

Reviewer 1

However, in my humble opinion the innovation continues being unclear to me. For me be the first and only system that incorporates ICF in a EHR is not enough to describe the real advances that your system provides, (that probably are a lot). I consider the authors can improve this part

Response: Thank you for this comment. Yes, the integration of a standardized process and terminology has many advantages. We added/corrected in the discussion first paragraph:

The implementation of the DCP and standardized terminology in HIS will support the documentation of the individual care process and thus quality assurance, e.g. transparency and continuity of care, on the one hand and facilitate the comparison of data and thus outcomes management and research, on the other hand [35,3]. As a result, best practice methods could be established and the effectiveness of care improved [3]. Moreover, no study was found in the literature that describes the analysis and assessment of the integration of a therapeutic process with standardized terminology. Thus, our detailed description, assessment and proposals for improvement can assist others in integrating the DCP and the ICF-Dietetics in their HIS. Our study can therefore be of interest to other institutions in Austria as well as in other countries.

Reviewer 2 Report

I notice that the paper has undertaken several improvements. In the results, potential levers and barriers to the implementation of the tool have been discussed.

However, I have some doubt that need to be cleared up about the methodology used in the paper. I noticed that the authors have modified the methodology section inserting the following text: "Four interviews were conducted by a computer scientist with an academic dietitian recruited from the developers of the DCP. These interviews were part of an interactive development process whereby feedback from other developers of the DCP and the interviewer also played a part. The interviews were structured around the five sub-processes of the DCP (dietetics assessment, dietetics diagnosis, dietetics goal setting, dietetics intervention and dietetics evaluation".

In the light of the iterative process that is described, I was wondering if the authors had played an active role in the development of the project or not.

Do the authors of this study have contributed to the development of the tool? If yes, the authors should insert a reference to action research in the methodology. If not, I suggest to insert details about the lenght of interviews, how coding around the main themes have been performed, how many feedbacks have been collected from the developers of the tool. This methodological process should be also supported with references.

Author Response

Reviewer 2

I notice that the paper has undertaken several improvements. In the results, potential levers and barriers to the implementation of the tool have been discussed.

However, I have some doubt that need to be cleared up about the methodology used in the paper. I noticed that the authors have modified the methodology section inserting the following text: "Four interviews were conducted by a computer scientist with an academic dietitian recruited from the developers of the DCP. These interviews were part of an interactive development process whereby feedback from other developers of the DCP and the interviewer also played a part. The interviews were structured around the five sub-processes of the DCP (dietetics assessment, dietetics diagnosis, dietetics goal setting, dietetics intervention and dietetics evaluation".

In the light of the iterative process that is described, I was wondering if the authors had played an active role in the development of the project or not.

Do the authors of this study have contributed to the development of the tool? If yes, the authors should insert a reference to action research in the methodology. If not, I suggest to insert details about the lenght of interviews, how coding around the main themes have been performed, how many feedbacks have been collected from the developers of the tool. This methodological process should be also supported with references.

Response: Thank you for this comment. Yes, it was part of an action research and an author contributed to the development process. Therefore, we added in the method section, first paragraph:

Using an action research approach [26-28] with qualitative semi-structured interviews, to identify strengths and weaknesses of the system and process integration.

AND

These interviews were part of an iterative development process, whereby the four typical development phases of planning, doing, checking and acting (PDCA-cycle) were performed several times so that results were analysed continuously, in order to improve quality of the data [26,27].

And we adapted the abstract as following:

Using an action research approach, the DCP was analysed through four expert interviews, and the integration into the HIS through two expert interviews with observations.

 To make this also in the introduction clearer, we added page 2, line 70: Beginning an action research process, the ICF-Dietetics was compared with the NCPT, translated, validated and clinically pre-tested to develop an Austrian-German ICF-Dietetics version [16], document templates were developed alongside this for the DCP, based on ICF documentation tools [17,18]. A multicentre study [14] and a focus group study [19] showed that the integration of the ICF-Dietetics into the DCP was possible and an Austria-wide implementation of the ICF-Dietetics was advocated. However, a multifaceted implementation strategy is required, with the integration of ICF-Dietetics into existing health information systems (HIS) seen as a prerequisite for a successful implementation [19].

Reviewer 3 Report

Authors are suggested to address the following comments.

Comment 1. Abstract:

(a) Organize the contents in single paragraph.

(b) Briefly share the key findings/results of the proposed work.

Comment 2. Some wordings/symbols are linked without spacing. Please confirm.

Comment 3. Section 1 Introduction:

(a) It seems that the paper was prepared some days ago, please update the list of references and focus on recently published journal articles.

(b) Literature review, it is suggested to summarize the methodology, results, and limitations of the existing works.

(c) Share the research contributions of the paper.

Comment 4. Section 2 Materials and Methods:

(a) Enhance the resolutions of all papers.

(b) Elaborate each of the system block in Figures 1 and 2.

Comment 5. Section 3 Results:

(a) Formal research analysis is missing.

(b) Performance comparison between proposed work and existing works is missing.

Author Response

Reviewer 3

Comment 1. Abstract:

(a) Organize the contents in single paragraph.

Response: Thank you for this comment. However, in the structure of the abstract we followed the guidelines of the journal.

(b) Briefly share the key findings/results of the proposed work.

Response: Thank you for this comment. We adapted the abstract:

Using an action research approach, the DCP was analysed through four expert interviews, and the integration into the HIS through two expert interviews with observations. Key strengths and weakness for the main criteria (“integration of the ICF catalogue”, “adaption of the document templates”, “adaption of the DCP” and the “adaption of the user authorizations”) were presented and proposals for improvement given. The system and process integration of DCP is possible and the document templates can be adapted with the software currently in use. Although an increase in resources and finances is to be expected initially, the integration of a standardized dietetic terminology in combination with a standardized process is likely to improve the quality of care and support outcomes management and research.

Comment 2. Some wordings/symbols are linked without spacing. Please confirm.

Response: Thank you for this comment. Unfortunately, there were formatting problems in the world-file with track changes, which we have now corrected.

Comment 3. Section 1 Introduction:

(a) It seems that the paper was prepared some days ago, please update the list of references and focus on recently published journal articles.

Response: Thank you for this comment. We checked the references and updated to more recent ones were available (see below), however the older ones remain as they are still relevant and necessary to the narrative of the article.

Iso/iec 15504-5:2012. ISO

To

  1. ISO. Iso/iec ts 33061:2021 information technology — process assessment — process assessment model for software life cycle processes. In Information technology — Process assessment — Process assessment model for software life cycle processes.

And we added new recent references of article:

  1. Sholle, E.T.; Cusick, M.; Davila, M.A.; Kabariti, J.; Flores, S.; Campion, T.R. Characterizing basic and complex usage of i2b2 at an academic medical center. AMIA Jt Summits Transl Sci Proc 2020, 2020, 589-596.
  2. Kuo, K.M.; Liu, C.F.; Talley, P.C.; Pan, S.Y. Strategic improvement for quality and satisfaction of hospital information systems. J Healthc Eng 2018, 2018, 3689618

(b) Literature review, it is suggested to summarize the methodology, results, and limitations of the existing works.

Response: Thank you for this comment. We did a literature review. Our system and process analysis and assessment however, build on previous studies, in accordance to an action research approach. Therefore, in our introduction we have focused on the work done so far and identified the gap which our study addresses. We have structured the introduction so that the reader can follow the entire research process.

We covered the following points:

The use of the ICF is recommended in combination with the already implemented ICD by the WHO.  However, the ICF is lacking of dietetics categories and therefore the ICF-Dietetics was developed in the Netherlands. 

The integration of ICF-Dietetics and other standardized terminologies into existing therapeutic process of health information systems was seen as a prerequisite for a successful implementation nationwide, among other things. That was the gap which our study addressed.

 (c) Share the research contributions of the paper.

Response: Our work of this actual paper produced a unique contribution to the field as we address this gap through the action research process: The ICF-Dietetics was compared with the second worldwide used dietetics language, translated into German and validated. A focus group study identified barriers and facilitators in terms of feasibility and acceptability. Our findings therefore pave the way for the integration of ICF-Dietetics on an international level.

Comment 4. Section 2 Materials and Methods:

(a) Enhance the resolutions of all papers.

Response: Thank you for this comment. This will be addressed during the editing process where we will submit a jpg graphic with a resolution of 600x600 for integration.

(b) Elaborate each of the system block in Figures 1 and 2.

Response: Based on the requests from previous reviews we shortened our article to make it more concise and additional detail is contained in supplementary material. Therefore, we have described the process steps and the document templates (Fig. 1) in detail in Supplement materials S1 and the sub-processes of the current process (Fig. 2) in S2.

 To make this clearer in the paragraph, we changed it in line 130-134:

The description of the new DCP with reference to the document templates and the standardized terminology (classification catalogues) is depicted in a simplified process diagram in Figure 1. This is described in detail, with respect to its five sub-processes, in Supplementary Material S1.

Comment 5. Section 3 Results:

(a) Formal research analysis is missing.

Response: To clarify our methods we have added:

Using an action research approach [26-28] with qualitative semi-structured interviews, to identify strengths and weaknesses of the system and process integration.

AND

These interviews were part of an iterative development process, whereby the four typical development phases of planning, doing, checking and acting (PDCA-cycle) were performed several times so that results were analysed continuously, in order to improve quality of the data [26,27].

 (b) Performance comparison between proposed work and existing works is missing.

Response: Thank you for this comment. Our study focused on the integration into a specific HIS as we wrote in the introduction (line 91-95):

In hospitals, HIS are used by different providers, have different requirements and have to meet different challenges. This heterogeneity makes it necessary for this system analysis and assessment to be carried out in the context of implementation projects on the basis of a specific HIS. Principles from learning here can be applied to other settings.

And line 89-91

This allows obstacles and potential for improvement to be identified, timely adaptations and, if necessary, alternative proposal to be made.

The Summary of main results in regard to our four criteria for the system and process integration: “integration of the classification catalogues”, “adaption of the document templates”, “adaption of the new DCP” and “user authorizations” we show in Table 1.

To point out this aspect, we added/corrected in the discussion some aspects:

This work demonstrates a system and process analysis and assessment for the integration of a new therapeutic process, the DCP, with a standardized terminology and document templates in an existing HIS. It was an action research approach, building on previous studies. The implementation of the DCP and standardized terminology in HIS will support the documentation of the individual care process and thus quality assurance, e.g. transparency and continuity of care, on the one hand and facilitate the comparison of data and thus outcomes management and research, on the other hand [35]. As a result, best practice methods could be established and the effectiveness of care improved [3]. Moreover, no study was found in the literature that describes the analysis and assessment of the integration of a therapeutic process with standardized terminology. Thus, our detailed description, assessment and proposals for improvement can assist others in integrating the DCP and the ICF-Dietetics in their HIS. Our study can therefore be of interest to other institutions in Austria as well as in other countries.

However, we want to emphasize that the change of a process must also take into account the non-technical aspects. During an information system analysis and assessment, it is important that the people involved in the process with their roles and skills are considered accordingly. Therefore, established process models for system analysis and assessment use two dimensions to categorize the areas of the criteria, namely organizational aspects in addition to the technical aspects of information tools [22,24]. Our system and process analysis and assessment build on a previous pretest and focus group study, which evaluated the clinical practicability and applicability of the new DCP with the ICF-Dietetics and the templates as ’paper and pencil’ version [19]. For this reason, we focused our findings to four main criteria which are important for the integration of a new process with documents and standardized terminologies: “integration of the classifications catalogue”; “adaption of the document templates”; “adaption of the new DCP” and the “adaption of the user authorizations”. Our results showed strengths and weaknesses in each of these areas, these were consistent with other literature as described below, and obstacles considered surmountable. In accordance with action research [26,27] the implementation of the new process will be evaluated in an implementation project and will be described elsewhere.

 Furthermore, we added to the limitation: Finally, the fact that no evaluation of the implementation of the new DCP was included can be seen as a limitation of this study. In this context, we would like to emphasize our methodological approach in the sense of action research, which enables a step by step translation and implementation of knowledge into practice.

Round 2

Reviewer 3 Report

Reviewer 3

Comment 1. Abstract:

(a) Organize the contents in single paragraph.

Response: Thank you for this comment. However, in the structure of the abstract we followed the guidelines of the journal.

Follow-up comment: Refer to the template: Abstract: A single paragraph of about 200 words maximum.

Comment 3. Section 1 Introduction:

(a) It seems that the paper was prepared some days ago, please update the list of references and focus on recently published journal articles.

Response: Thank you for this comment. We checked the references and updated to more recent ones were available (see below), however the older ones remain as they are still relevant and necessary to the narrative of the article.

Follow-up comment: Many new articles have been included. However, the major references covered in current article are not latest works. Particularly, cover articles from 2020.

(b) Literature review, it is suggested to summarize the methodology, results, and limitations of the existing works.

Response: Thank you for this comment. We did a literature review. Our system and process analysis and assessment however, build on previous studies, in accordance to an action research approach. Therefore, in our introduction we have focused on the work done so far and identified the gap which our study addresses. We have structured the introduction so that the reader can follow the entire research process.

We covered the following points:

The use of the ICF is recommended in combination with the already implemented ICD by the WHO.  However, the ICF is lacking of dietetics categories and therefore the ICF-Dietetics was developed in the Netherlands. 

The integration of ICF-Dietetics and other standardized terminologies into existing therapeutic process of health information systems was seen as a prerequisite for a successful implementation nationwide, among other things. That was the gap which our study addressed.

Follow-up comment: Shallow discussion was made on the literature review.

Comment 5. Section 3 Results:

(a) Formal research analysis is missing.

Response: To clarify our methods we have added:

Using an action research approach [26-28] with qualitative semi-structured interviews, to identify strengths and weaknesses of the system and process integration.

AND

These interviews were part of an iterative development process, whereby the four typical development phases of planning, doing, checking and acting (PDCA-cycle) were performed several times so that results were analysed continuously, in order to improve quality of the data [26,27].

Follow-up comment: Elaborate the data collection process (including the the design) of the interviews. 

(b) Performance comparison between proposed work and existing works is missing.

Response: Thank you for this comment. Our study focused on the integration into a specific HIS as we wrote in the introduction (line 91-95):

In hospitals, HIS are used by different providers, have different requirements and have to meet different challenges. This heterogeneity makes it necessary for this system analysis and assessment to be carried out in the context of implementation projects on the basis of a specific HIS. Principles from learning here can be applied to other settings.

And line 89-91

This allows obstacles and potential for improvement to be identified, timely adaptations and, if necessary, alternative proposal to be made.

The Summary of main results in regard to our four criteria for the system and process integration: “integration of the classification catalogues”, “adaption of the document templates”, “adaption of the new DCP” and “user authorizations” we show in Table 1.

To point out this aspect, we added/corrected in the discussion some aspects:

This work demonstrates a system and process analysis and assessment for the integration of a new therapeutic process, the DCP, with a standardized terminology and document templates in an existing HIS. It was an action research approach, building on previous studies. The implementation of the DCP and standardized terminology in HIS will support the documentation of the individual care process and thus quality assurance, e.g. transparency and continuity of care, on the one hand and facilitate the comparison of data and thus outcomes management and research, on the other hand [35]. As a result, best practice methods could be established and the effectiveness of care improved [3]. Moreover, no study was found in the literature that describes the analysis and assessment of the integration of a therapeutic process with standardized terminology. Thus, our detailed description, assessment and proposals for improvement can assist others in integrating the DCP and the ICF-Dietetics in their HIS. Our study can therefore be of interest to other institutions in Austria as well as in other countries.

However, we want to emphasize that the change of a process must also take into account the non-technical aspects. During an information system analysis and assessment, it is important that the people involved in the process with their roles and skills are considered accordingly. Therefore, established process models for system analysis and assessment use two dimensions to categorize the areas of the criteria, namely organizational aspects in addition to the technical aspects of information tools [22,24]. Our system and process analysis and assessment build on a previous pretest and focus group study, which evaluated the clinical practicability and applicability of the new DCP with the ICF-Dietetics and the templates as ’paper and pencil’ version [19]. For this reason, we focused our findings to four main criteria which are important for the integration of a new process with documents and standardized terminologies: “integration of the classifications catalogue”; “adaption of the document templates”; “adaption of the new DCP” and the “adaption of the user authorizations”. Our results showed strengths and weaknesses in each of these areas, these were consistent with other literature as described below, and obstacles considered surmountable. In accordance with action research [26,27] the implementation of the new process will be evaluated in an implementation project and will be described elsewhere.

 Furthermore, we added to the limitation: Finally, the fact that no evaluation of the implementation of the new DCP was included can be seen as a limitation of this study. In this context, we would like to emphasize our methodological approach in the sense of action research, which enables a step by step translation and implementation of knowledge into practice.

Follow-up comment: What are the research contributions in terms of methodology and results given that no formal analysis and evaluation was made?

Author Response

Reviewer 3

Comment 1. Abstract:

(a) Organize the contents in single paragraph.

Response: Thank you for this comment. However, in the structure of the abstract we followed the guidelines of the journal.

Follow-up comment 1: Refer to the template: Abstract: A single paragraph of about 200 words maximum.

Response to Follow-up comment 1: We followed the guidelines of the journal: (https://www.mdpi.com/authors/layout#_bookmark5)

2.4. Abstracts
The abstract contains a summary of the entire paper and can be up to 200 words long with only one paragraph.

And we based the abstract on the IMRAD structure of a paper but without using headings.

Comment 3. Section 1 Introduction:

(a) It seems that the paper was prepared some days ago, please update the list of references and focus on recently published journal articles.

Response: Thank you for this comment. We checked the references and updated to more recent ones were available (see below), however the older ones remain as they are still relevant and necessary to the narrative of the article.

Follow-up comment 3: Many new articles have been included. However, the major references covered in current article are not latest works. Particularly, cover articles from 2020.

Response to Follow-up comment 3a: Thank you for this comment. We integrated in the indroduction a new paragraph to cover many new ICF publications with some older ones considering the same topic:

Numerous articles can be found regarding ICF. These describe how the ICF framework can be used in multidisciplinary healthcare [13-18], discuss ICF Core Set (sets of ICF categories relevant for patients with a certain heath condition), describe development and validation to facilitate multidisciplinary assessment [19-27]. Articles compare the content of instruments to measure functioning of patients [28-35] and for the linking of problems experienced from a patient perspective in daily life [36-38]. Furthermore, studies describe the implementation of ICF-based tools in clinical practice [39-46] and about ICF use in electronic health records (EHRs) [47,48].

Rauch, A.; Cieza, A.; Stucki, G. How to apply the International Classification of Functioning, Disability and Health (ICF) for rehabilitation management in clinical practice. Eur J Phys Rehabil Med 2008, 44, 329-342.

  1. Kotsougiani-Fischer, D.; Choi, J.S.; Oh-Fischer, J.S.; Diehm, Y.F.; Haug, V.F.; Harhaus, L.; Gazyakan, E.; Hirche, C.; Kneser, U.; Fischer, S. ICF-based multidisciplinary rehabilitation program for complex regional pain syndrome of the hand: efficacy, long-term outcomes, and impact of therapy duration. BMC Surg 2020, 20, 306, doi:10.1186/s12893-020-00982-7.
  2. Zhang, M.; Zhang, Y.; Xiang, Y.; Lin, Z.; Shen, W.; Wang, Y.; Wang, L.; Yu, J.; Yan, T. A team approach to applying the International Classification of Functioning, Disability and Health Rehabilitation set in clinical evaluation. J Rehabil Med 2021, 53, jrm00147, doi:10.2340/16501977-2756.
  3. Stucki, G.; Pollock, A.; Engkasan, J.P.; Selb, M. How to use the International Classification of Functioning, Disability and Health as a reference system for comparative evaluation and standardized reporting of rehabilitation interventions. Eur J Phys Rehabil Med 2019, 55, 384-394, doi:10.23736/S1973-9087.19.05808-8.
  4. Brunani, A.; Raggi, A.; Sirtori, A.; Berselli, M.E.; Villa, V.; Ceriani, F.; Corti, S.; Leonardi, M.; Capodaglio, P.; Group, I.-O. An ICF-Based Model for Implementing and Standardizing Multidisciplinary Obesity Rehabilitation Programs within the Healthcare System. Int J Environ Res Public Health 2015, 12, 6084-6091, doi:10.3390/ijerph120606084.
  5. Brunani, A.; Sirtori, A.; Capodaglio, P.; Donini, L.M.; Buscemi, S.; Carbonelli, M.G.; Giordano, F.; Mazzali, G.; Pasqualinotto, L.; Zenti, M.G.; et al. Disability assessment in an Italian cohort of patients with obesity using an International Classification of Functioning, Disability and Health (ICF)-derived questionnaire. Eur J Phys Rehabil Med 2021, 57, 630-638, doi:10.23736/S1973-9087.20.06512-0.
  6. Selb, M.; Escorpizo, R.; Kostanjsek, N.; Stucki, G.; Ustun, B.; Cieza, A. A guide on how to develop an International Classification of Functioning, Disability and Health Core Set. Eur J Phys Rehabil Med 2015, 51, 105-117.
  7. Zangger, M.; Weber, C.; Stute, P. Developing an ICF Core Set for Climacteric Syndrome based on the International Classification of Functioning, Disability and Health (ICF). Maturitas 2021, 143, 197-202, doi:10.1016/j.maturitas.2020.10.014.
  8. Tomandl, J.; Book, S.; Hoefle, A.; Graessel, E.; Sieber, C.; Freiberger, E.; Kuehlein, T.; Hueber, S.; Gotthardt, S. Laying the foundation for a primary care core set of the International Classification of Functioning, Disability and Health (ICF) for community-dwelling older adults: A qualitative study. J Rehabil Med 2021, 53, jrm00150, doi:10.2340/16501977-2779.
  9. Stucki, A.; Cieza, A.; Michel, F.; Stucki, G.; Bentley, A.; Culebras, A.; Tufik, S.; Kotchabhakdi, N.; Tachibana, N.; Ustun, B.; et al. Developing ICF Core Sets for persons with sleep disorders based on the International Classification of Functioning, Disability and Health. Sleep Med 2008, 9, 191-198, doi:10.1016/j.sleep.2007.01.019.
  10. Stucki, A.; Daansen, P.; Fuessl, M.; Cieza, A.; Huber, E.; Atkinson, R.; Kostanjsek, N.; Stucki, G.; Ruof, J. ICF Core Sets for obesity. J Rehabil Med 2004, 107-113, doi:10.1080/16501960410016064.
  11. Stucki, G.; Cieza, A.; Geyh, S.; Battistella, L.; Lloyd, J.; Symmons, D.; Kostanjsek, N.; Schouten, J. ICF Core Sets for rheumatoid arthritis. J Rehabil Med 2004, 44, 87-93, doi:10.1080/16501960410015470.
  12. Tschiesner, U.; Linseisen, E.; Becker, S.; Mast, G.; Rogers, S.N.; Walvekar, R.R.; Berghaus, A.; Cieza, A. Content validation of the international classification of functioning, disability and health core sets for head and neck cancer: a multicentre study. J Otolaryngol Head Neck Surg 2010, 39, 674-687.
  13. Viehoff, P.B.; Potijk, F.; Damstra, R.J.; Heerkens, Y.F.; van Ravensberg, C.D.; van Berkel, D.M.; Neumann, H.A. Identification of relevant ICF (International Classification of Functioning, Disability and Health) categories in lymphedema patients: A cross-sectional study. Acta Oncol 2015, 54, 1218-1224, doi:10.3109/0284186X.2014.1001873.
  14. Xie, F.; Lo, N.N.; Lee, H.P.; Cieza, A.; Li, S.C. Validation of the International Classification of Functioning, Disability, and Health (ICF) Brief Core Set for osteoarthritis. Scand J Rheumatol 2008, 37, 450-461, doi:10.1080/03009740802116216.
  15. Coenen, M.; Kus, S.; Rudolf, K.-D.; Müller, G.; Berno, S.; Dereskewitz, C.; MacDermid, J. Do patient-reported outcome measures capture functioning aspects and environmental factors important to individuals with injuries or disorders of the hand? J Hand Ther 2013, 26, 332-342; quiz 342, doi:10.1016/j.jht.2013.06.002.
  16. D'Amico, D.; Tepper, S.J.; Guastafierro, E.; Toppo, C.; Leonardi, M.; Grazzi, L.; Martelletti, P.; Raggi, A. Mapping Assessments Instruments for Headache Disorders against the ICF Biopsychosocial Model of Health and Disability. Int J Environ Res Public Health 2020, 18, doi:10.3390/ijerph18010246.
  17. Alghwiri, A.A.; Almhdawi, K.A.; Marchetti, G. Are fatigue scales the same? A content comparison using the International Classification of Functioning, Disability and Health. Mult Scler Relat Disord 2020, 46, 102596, doi:10.1016/j.msard.2020.102596.
  18. Cieza, A.; Stucki, G. Content comparison of health-related quality of life (HRQOL) instruments based on the international classification of functioning, disability and health (ICF). Qual Life Res 2005, 14, 1225-1237.
  19. Prodinger, B.; Cieza, A.; Williams, D.A.; Mease, P.; Boonen, A.; Kerschan-Schindl, K.; Fialka-Moser, V.; Smolen, J.; Stucki, G.; Machold, K.; et al. Measuring health in patients with fibromyalgia: content comparison of questionnaires based on the International Classification of Functioning, Disability and Health. Arthritis Rheum 2008, 59, 650-658, doi:10.1002/art.23559.
  20. Stamm, T.A.; Cieza, A.; Machold, K.P.; Smolen, J.S.; Stucki, G. Content comparison of occupation-based instruments in adult rheumatology and musculoskeletal rehabilitation based on the International Classification of Functioning, Disability and Health. Arthritis Rheum 2004, 51, 917-924, doi:10.1002/art.20842.
  21. Stucki, A.; Borchers, M.; Stucki, G.; Cieza, A.; Amann, E.; Ruof, J. Content comparison of health status measures for obesity based on the international classification of functioning, disability and health. International Journal of Obesity 2006, 30, 1791-1799, doi:10.1038/sj.ijo.0803335.
  22. Patel, K.; Straudi, S.; Yee Sien, N.; Fayed, N.; Melvin, J.L.; Sivan, M. Applying the WHO ICF Framework to the Outcome Measures Used in the Evaluation of Long-Term Clinical Outcomes in Coronavirus Outbreaks. Int J Environ Res Public Health 2020, 17, doi:10.3390/ijerph17186476.
  23. Cairns, I.; Lindsay, K.; Dalbeth, N.; Diaz-Torne, C.; Antonia Pou, M.; Rodriguez Diez, B.; Pujol-Ribera, E.; Panter, C.; Arbuckle, R.; Tatlock, S.; et al. The impact of gout as described by patients, using the lens of The International Classification of Functioning, Disability and Health (ICF): a qualitative study. BMC Rheumatol 2020, 4, 50, doi:10.1186/s41927-020-00147-2.
  24. Coenen, M.; Cabello, M.; Umlauf, S.; Ayuso-Mateos, J.L.; Anczewska, M.; Tourunen, J.; Leonardi, M.; Cieza, A.; Consortium, P. Psychosocial difficulties from the perspective of persons with neuropsychiatric disorders. Disabil Rehabil 2016, 38, 1134-1145, doi:10.3109/09638288.2015.1074729.
  25. Stamm, T.A.; Machold, K.; Sahinbegovic, E.; Haider, S.; Ernst, M.; Binder, A.; Dallos, T.; Zwerina, J.; Smolen, J. Daily functioning and health status in patients with hand osteoarthritis: Fewer differences between women and men than expected. Wien Klin Wochenschr 2011, 123, 603-606, doi:10.1007/s00508-011-1597-0.
  26. van Leeuwen, L.M.; Pronk, M.; Merkus, P.; Goverts, S.T.; Anema, J.R.; Kramer, S.E. Developing an intervention to implement an ICF-based e-intake tool in clinical otology and audiology practice. Int J Audiol 2020, 59, 282-300, doi:10.1080/14992027.2019.1691746.
  27. van Leeuwen, L.M.; Pronk, M.; Merkus, P.; Goverts, S.T.; Terwee, C.B.; Kramer, S.E. Operationalization of the Brief ICF Core Set for Hearing Loss: An ICF-Based e-Intake Tool in Clinical Otology and Audiology Practice. Ear Hear 2020, 41, 1533-1544, doi:10.1097/AUD.0000000000000867.
  28. Appleby, H.; Tempest, S. Using change management theory to implement the International Classification of Functioning, Disability and Health (ICF) in clinical practice. British Journal of Occupational Therapy 2006, 69, 477-480.
  29. Tempest, S.; Harries, P.; Kilbride, C.; De Souza, L. Enhanced clarity and holism: the outcome of implementing the ICF with an acute stroke multidisciplinary team in England. Disabil Rehabil 2013, 35, 1921-1925, doi:10.3109/09638288.2013.766272.
  30. Tempest, S.; Harries, P.; Kilbride, C.; De Souza, L. To adopt is to adapt: the process of implementing the ICF with an acute stroke multidisciplinary team in England. Disabil Rehabil 2012, 34, 1686-1694, doi:10.3109/09638288.2012.658489.
  31. Tempest, S.; Jefferson, R. Engaging with clinicians to implement and evaluate the ICF in neurorehabilitation practice. NeuroRehabilitation 2015, 36, 11-15, doi:10.3233/NRE-141185.
  32. Mukaino, M.; Prodinger, B.; Yamada, S.; Senju, Y.; Izumi, S.I.; Sonoda, S.; Selb, M.; Saitoh, E.; Stucki, G. Supporting the clinical use of the ICF in Japan - development of the Japanese version of the simple, intuitive descriptions for the ICF Generic-30 set, its operationalization through a rating reference guide, and interrater reliability study. BMC Health Serv Res 2020, 20, 66, doi:10.1186/s12913-020-4911-6.
  33. Coenen, M.; Rudolf, K.D.; Kus, S.; Dereskewitz, C. [The International Classification of Functioning, Disability and Health (ICF) : The implementation of the ICF Core Sets for Hand Conditions in clinical routine as an example of application]. Bundesgesundheitsblatt Gesundheitsforschung Gesundheitsschutz 2018, 61, 787-795, doi:10.1007/s00103-018-2748-5.
  34. Maritz, R.; Aronsky, D.; Prodinger, B. The International Classification of Functioning, Disability and Health (ICF) in Electronic Health Records. A Systematic Literature Review. Appl Clin Inform 2017, 8, 964-980, doi:10.4338/ACI-2017050078.
  35. Cozzi, S.; Martinuzzi, A.; Della Mea, V. Ontological modeling of the International Classification of Functioning, Disabilities and Health (ICF): Activities&Participation and Environmental Factors components. BMC Med Inform Decis Mak 2021, 21, 367, doi:10.1186/s12911-021-01729-x.

(b) Literature review, it is suggested to summarize the methodology, results, and limitations of the existing works.

Response: Thank you for this comment. We did a literature review. Our system and process analysis and assessment however, build on previous studies, in accordance to an action research approach. Therefore, in our introduction we have focused on the work done so far and identified the gap which our study addresses. We have structured the introduction so that the reader can follow the entire research process.

We covered the following points:

The use of the ICF is recommended in combination with the already implemented ICD by the WHO.  However, the ICF is lacking of dietetics categories and therefore the ICF-Dietetics was developed in the Netherlands. 

The integration of ICF-Dietetics and other standardized terminologies into existing therapeutic process of health information systems was seen as a prerequisite for a successful implementation nationwide, among other things. That was the gap which our study addressed.

Follow-up comment: Shallow discussion was made on the literature review.

Response to Follow-up comment 3b: See our additional paragraph above in response to comment 3.

Comment 5. Section 3 Results:

(a) Formal research analysis is missing.

Response: To clarify our methods we have added:

Using an action research approach [26-28] with qualitative semi-structured interviews, to identify strengths and weaknesses of the system and process integration.

AND

These interviews were part of an iterative development process, whereby the four typical development phases of planning, doing, checking and acting (PDCA-cycle) were performed several times so that results were analysed continuously, in order to improve quality of the data [26,27].

Follow-up comment: Elaborate the data collection process (including the design) of the interviews. 

Response to Follow-up comment 5a: To make this clearer, we changed our data collection description as followed:

Four interviews were conducted by a computer scientist (principal investigator) with an academic dietitian recruited from the developers of the DCP. The interviews were structured around the five sub-processes of the DCP (dietetics assessment, dietetics diagnosis, dietetics goal setting, dietetics intervention and dietetics evaluation). These interviews were part of an iterative development process whereby the four typical development phases of planning, doing, checking, and acting (PDCA-cycle) were performed several times in order to improve the quality of the analysis results continuously [60,61]. Feedback from other developers of the DCP and the interviewer also played a part. During this process, graphical representations of the new DCP and a precise description for the analysis and assessment with respect to its five sub-processes were carried out by the principal investigator. The purpose of these was to develop, test and refine the process through discussion of what works well and what barriers arise. The graphical representation of the new DCP with reference to the document templates and the standardized terminology (classification catalogues) was depicted in a simplified process diagram ( Figure 1). This is described in detail in Supplementary Material S1. (The referral required according to § 2 of the MTD law [49] in case of a medical diagnosis is not shown explicitly in Figure 1, since the focus is on the implementation of the new DCP by dietitians and this starts with the assessment. Expert systems and clinical decision support were hypothetically included in the interviews (Supplementary Material S1), however, were not assessed).

AND for the current process:

Two further interviews, together with on-site observations of use of the current process were conducted by the principal investigator with the leading dietitian of the hospital. These interviews also formed part of an interactive process with feedback from other dietitians of the hospital. During this development process, the current process was visualized as a business process model ("extended event-driven process chain" (eEPC)) and used as a framework for assessing the integration of the new DCP within it. The current process is divided into four sub-processes due to its complexity (Figure 2). The whole business process model is shown and described in Supplementary Material S2. The following specific requirements were applied to the modelling language of the current process: 1. Trivial notation elements for users without modelling expertise, kept as general as possible and represent-able without additional specifics or explanations, 2. Modelling language close to the existing Enterprise Resource Planning (ERP) software SAP, and standardized modelling language that enables processing by external persons. The software "EdrawMax" [65] was used for modelling

Qualitative data of the assessment was mapped against four main criteria which are important for the integration of a new process with documents and standardized terminologies: for the system integration, the “integration of the ICF catalogue” and the “adaption of the document templates”, and for the process integration, the “adaption of the new DCP” and the “adaption of the user authorizations.

(b) Performance comparison between proposed work and existing works is missing.

Response: Thank you for this comment. Our study focused on the integration into a specific HIS as we wrote in the introduction (line 91-95):

In hospitals, HIS are used by different providers, have different requirements and have to meet different challenges. This heterogeneity makes it necessary for this system analysis and assessment to be carried out in the context of implementation projects on the basis of a specific HIS. Principles from learning here can be applied to other settings.

And line 89-91

This allows obstacles and potential for improvement to be identified, timely adaptations and, if necessary, alternative proposal to be made.

The Summary of main results in regard to our four criteria for the system and process integration: “integration of the classification catalogues”, “adaption of the document templates”, “adaption of the new DCP” and “user authorizations” we show in Table 1.

To point out this aspect, we added/corrected in the discussion some aspects:

This work demonstrates a system and process analysis and assessment for the integration of a new therapeutic process, the DCP, with a standardized terminology and document templates in an existing HIS. It was an action research approach, building on previous studies. The implementation of the DCP and standardized terminology in HIS will support the documentation of the individual care process and thus quality assurance, e.g. transparency and continuity of care, on the one hand and facilitate the comparison of data and thus outcomes management and research, on the other hand [35]. As a result, best practice methods could be established and the effectiveness of care improved [3]. Moreover, no study was found in the literature that describes the analysis and assessment of the integration of a therapeutic process with standardized terminology. Thus, our detailed description, assessment and proposals for improvement can assist others in integrating the DCP and the ICF-Dietetics in their HIS. Our study can therefore be of interest to other institutions in Austria as well as in other countries.

However, we want to emphasize that the change of a process must also take into account the non-technical aspects. During an information system analysis and assessment, it is important that the people involved in the process with their roles and skills are considered accordingly. Therefore, established process models for system analysis and assessment use two dimensions to categorize the areas of the criteria, namely organizational aspects in addition to the technical aspects of information tools [22,24]. Our system and process analysis and assessment build on a previous pretest and focus group study, which evaluated the clinical practicability and applicability of the new DCP with the ICF-Dietetics and the templates as ’paper and pencil’ version [19]. For this reason, we focused our findings to four main criteria which are important for the integration of a new process with documents and standardized terminologies: “integration of the classifications catalogue”; “adaption of the document templates”; “adaption of the new DCP” and the “adaption of the user authorizations”. Our results showed strengths and weaknesses in each of these areas, these were consistent with other literature as described below, and obstacles considered surmountable. In accordance with action research [26,27] the implementation of the new process will be evaluated in an implementation project and will be described elsewhere.

 Furthermore, we added to the limitation: Finally, the fact that no evaluation of the implementation of the new DCP was included can be seen as a limitation of this study. In this context, we would like to emphasize our methodological approach in the sense of action research, which enables a step by step translation and implementation of knowledge into practice.

Follow-up comment: What are the research contributions in terms of methodology and results given that no formal analysis and evaluation was made?

Response to Follow-up comment 5b: We disagree – as a formal qualitative analysis was contacted using participatory research approach as described previously.

This manuscript is a resubmission of an earlier submission. The following is a list of the peer review reports and author responses from that submission.

Round 1

Reviewer 1 Report

In general, the article is well written and organized. I consider that the integration of standardized terminologies and document templates in a health information system for standardized documentation of the dietetic care process it is important and contributes to the quality improvement of clinical dietetic practice. Also, the development is justified due to there are currently no medical expert systems in use in the field of dietitians in Austria.  Moreover, the parts included in the document templates are adequate. However, I think it is too descriptive in terms of the documents developed, but not in terms of the interviews and discussions of the experts.
Finally, here are some formatting errors that could be changed:
Line 73- separate in two words “informationsystems”
Line 277- Add an “e” in the word Therefore

Reviewer 2 Report

The paper presents a methodology for standardize the documentation in dietetic care processes. 

The title of the paper is so long and difficult to follow. I suggest to change it to a more explainable/readable one.
The paper is unbalanced, the 90% is occuped by results and discussion. It is necessary to present a better related work in order to better highlight the innovation described by the paper. 

The paper seems more an implementation of a protocol than a research paper. I don't see the research. As an innovation paper,the authors said that this is a method never used on Dietetics field. In any case I consider mandatory to explain in a related work section how this method is explained in other fields. I consider the text included in the paper about other works inssuficient.

The Results are very well described and the discussion is very complete. 

Reviewer 3 Report

The manuscript aims to analyze and assess system and process technology the integration of the ICF-Dietetics. The study show a novel approach to improve a health information system for standardized documentation of the dietetic care process.

However I have some questions about the manuscript. First, I am unsure whether this manuscript nicely aligns with the journal's aims and scopes. This manuscript looks too technical and may have a limited interest of some specific health professionals. If not, this manuscript may need to discuss how it can contribute to public and environmental health in general.

Second, I am also concerned about how well research methods were designed for this project--L103-109 were unclear; for example., how the authors selected the interviewees? What were the interview questions? What standards were used for the assessment? etc.

Third, I am not sure which ones are analysis and assessment in the manuscript. 

I hope my comments may help to improve the manuscript.